# Electrically driven cation exchange for *in situ* fabrication of individual nanostructures

Qiubo Zhang[1,*], Kuibo Yin[1,*], Hui Dong[1], Yilong Zhou[1], Xiaodong Tan[1], Kaihao Yu[1], Xiaohui Hu[1,2], Tao Xu[1], Chao Zhu[1], Weiwei Xia[1], Feng Xu[1], Haimei Zheng[3,4] & Litao Sun[1,5]

Cation exchange (CE) has been recognized as a particularly powerful tool for the synthesis of heterogeneous nanocrystals. At present, CE can be divided into two categories, namely ion solvation-driven CE reaction and thermally activated CE reaction. Here we report an electrically driven CE reaction to prepare individual nanostructures inside a transmission electron microscope. During the process, Cd is eliminated due to Ohmic heating, whereas $Cu^+$ migrates into the crystal driven by the electrical field force. Contrast experiments reveal that the feasibility of electrically driven CE is determined by the structural similarity of the sulfur sublattices between the initial and final phases, and the standard electrode potentials of the active electrodes. Our experimental results demonstrate a strategy for the selective growth of individual nanocrystals and provide crucial insights into understanding of the microscopic pathways leading to the formation of heterogeneous structures.

[1] SEU-FEI Nano-Pico Center, Key Laboratory of MEMS of Ministry of Education, School of Electronic Science and Engineering, Southeast University, Nanjing 210018, China. [2] College of Materials Science and Engineering, Nanjing Tech University, Nanjing 210009, China. [3] Materials Science Division, Lawrence Berkeley National Laboratory, Berkeley, California 94720, USA. [4] Department of Materials Science and Engineering, University of California, Berkeley, California 94720, USA. [5] Center for Advanced Materials and Manufacture, Joint Research Institute of Southeast University and Monash University, Suzhou 215123, China. * These authors contributed equally to this work. Correspondence and requests for materials should be addressed to L.S. (email: slt@seu.edu.cn) or to H.Z. (email: hmzheng@lbl.gov).

Cation exchange (CE) reactions, in which one type of cation ligated within an intact anion sublattice are substituted by another kind of cation, have been regarded as a particularly powerful approach for the growth of heterogeneous structures that are not easily obtained using direct synthesis techniques[1–3]. The feasibility and the rates of CE processes depend both on the solubility product constant and the nature of the intervening activation barriers during reaction[4,5].

Depending on the nature of the strategy adopted to create a thermodynamic imbalance (TDI), CE methods can be divided into two categories[6]. One is based on CE activated by the ion solvation (Supplementary Fig. 1a), in which the TDI in the system is caused by the selective binding of the liquid ligand environment (LLE) to different sets of ions[4,7] and the effect of heat is to reduce the activation barriers[4]. Here, the direction and rates of CE can be controlled by adjusting the LLE and the reaction temperature[4]. However, the very dynamic nature of the liquid reaction environment makes it difficult to directly observe the evolution process with a high resolution[6]. In addition, due to the volume limitations in the LLE, it is not suitable for the selective preparation of individual nanostructures. The second category of CE is based on thermal activation (Supplementary Fig. 1b), in which the TDI in the reaction is caused by vacancies induced by evaporation[6,8–10]. The effect of heat is both to reduce the intervening activation barriers, as well as to cause the evaporation of ions[8]. This process can take place in the solid state, which allows us to easily monitor the CE process[6]. On the other hand, in such a system, it is difficult to selectively synthesize individual NCs due to the lack of appropriate sources for migrating ions. Furthermore, the thermally activated CE is random and uncontrollable, in that the free ingoing ion species randomly diffuse over the inert substrate. Therefore, up to now, this method has not been widely adopted.

Herein, we report an *in situ* electrically driven CE process for fabrication of an individual sulfide nanostructure inside a scanning tunnelling microscopic–transmission electron microscope (TEM) system, where the entire reaction process was monitored at the atomic scale in real time. Using this method (Supplementary Fig. 1c), it is possible to completely control the CE process and to selectively fabricate individual nanocrystals by applying an electric bias. Thus, it is possible to make more complicated nanostructures with controlled specific structures. We study the formation process and reaction dynamics of the individual heterogeneous nanostructures in detail.

## Results

**Construction of the experimental setup.** CdS nanowires (NWs) were synthesized by a previously reported solvothermal method[11] and were characterized by means of TEM, powder X-ray diffraction, energy-dispersive X-ray spectroscopy and electron energy-loss spectroscopy (EELS) (see Supplementary Fig. 2). The as-prepared CdS NWs have an average diameter of ~40 nm (Supplementary Fig. 2b,f) and were up to several micrometres long (Supplementary Fig. 2a). The NWs are single crystals with the wurtzite (WZ) structure, oriented in the <002> growth direction, as shown in Supplementary Fig. 2d,e. Supplementary Fig. 2g,h show that the as-prepared CdS NWs have no impurity element doping and the presence of Ni (Supplementary Fig. 2g) arises from the Ni support grid. The CdS NWs enable the facile evaporation of Cd by sublimation. Hence, we have chosen these NWs as precursors for the CE reaction[8–10].

The CE experiments were conducted inside a Cs-corrected TEM (FEI Titan 80–300). The schematic diagram of the *in situ* experimental setup integrated into the TEM is illustrated in Fig. 1a. On the left side, the chemically active Cu electrode, which

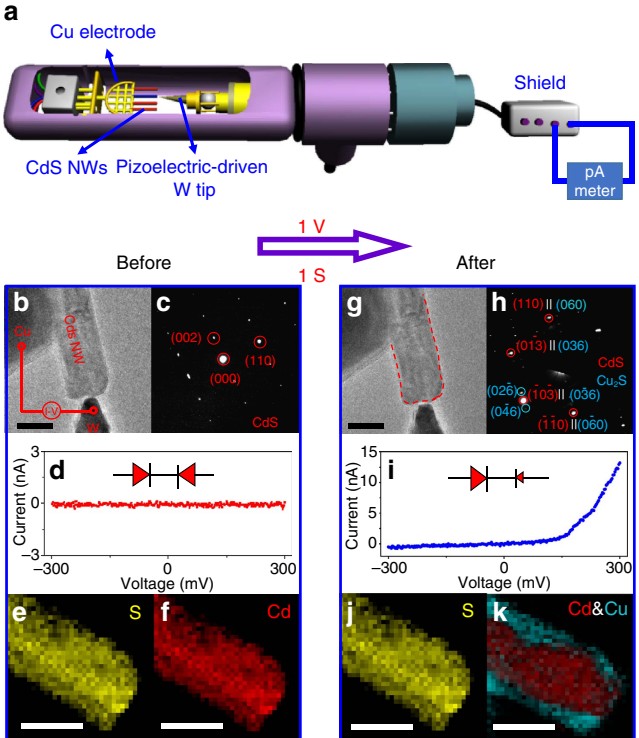

**Figure 1 | The formation of CdS/Cu₂S core-shell structure nanowire after bias in TEM. (a)** Schematic illustration of the *in situ* experimental setup. **(b–f)** Characterization of the nanowire (NW) before biasing. **(b)** Low-magnification TEM image of the experimental setup showing a CdS NW sandwiched between the Cu electrode and the W electrode. **(c)** SAED pattern showing the pristine CdS NW with hexagonal crystal structure. **(d)** Current–voltage (*I–V*) curve with the inset showing the equivalent circuit of the Cu-CdS-W structure; the formation of back-to-back Schottky contacts at both the interfaces is illustrated. **(e,f)** Corresponding STEM-EELS elemental mapping images for S **(e)** and Cd **(f)** show that S and Cd are distributed uniformly in the CdS NW. **(g–k)** Characterization of the NW after biasing. **(g)** Low-magnification TEM image of the NW after biasing and the red dotted line trace the boundary of the initial NW before biasing, from which it can be ascertained that no obvious shape change of the NW occurs after biasing. **(h)** SAED pattern of the NW indicates that it is composed of wurtzite CdS and LC Cu₂S phases. **(i)** *I–V* curve with the insert where the equivalent circuit of the Cu-Cu₂S/CdS-W structure having rectifying characteristics is shown. **(j,k)** Corresponding STEM-EELS elemental mapping images for S **(j)**, and Cd and Cu **(k)** of the NW after biasing confirms the CdS/Cu₂S core-shell heterostructure of the NW. Scale bars, 50 nm **(b,e,f,g,j, k)**.

supported the vertically oriented CdS NWs (diameters about 40 nm) and was the source of Cu cations[12], was glued to an Au rod using conductive epoxy (Chemtronics, CW2400). On the right side, a W tip was fixed onto a piezo manipulator and functioned as a movable and chemically inert electrode[12].

**Electrically driven CE.** During an *in situ* TEM experiment, the W tip was manipulated to touch one of the CdS NWs following which a positive voltage of 1.0 V was applied to the Cu electrode against the W tip for 1 s. The electron beam was blocked during biasing, to avoid the influence of the electron beam irradiation effects. Figure 1b–k show respectively, the characteristics of the NW before and after biasing. Figure 1b shows the low-magnification TEM image of the CdS NW before biasing. In this

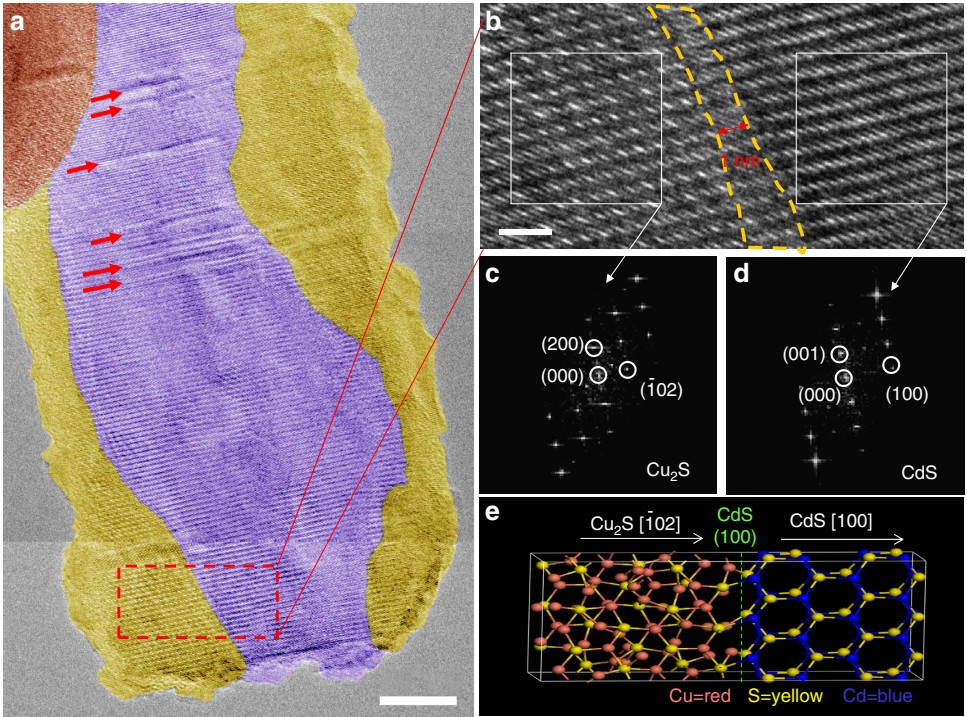

**Figure 2 | Detailed characterization of a CdS/Cu$_2$S core-shell structured NW.** (**a**) High-resolution TEM pseudo colour image of the CdS-Cu$_2$S heterostructure NW showing stacking faults in the CdS core indicated by red arrows. The purple area represents CdS core and the gold area represents Cu$_2$S shell. Scale bar, 10 nm. (**b**) High-resolution image of the region included by the red dotted square in **a**. The region under the yellow dotted line (the reaction zone) has a width of ∼1 nm. Scale bar, 2 nm. (**c,d**) The FFT images on the left-hand side (**c**) and the right-hand side (**d**) of the interface in the CdS/Cu$_2$S heterostructure at the Cu$_2$S $\{\overline{1}02\}/\{\overline{1}00\}$ CdS interface. (**e**) Schematic diagram of the CdS-Cu$_2$S heterostructure, corresponding to **b**.

image, the CdS NW is seen sandwiched between the Cu grid and the W tip, forming a metal–semiconductor–metal structure. Selected-area electron diffraction (SAED) pattern of a CdS NW (Fig. 1c) shows that the pristine CdS NW is a single crystal with WZ structure and EELS mapping (Fig. 1e,f) confirms that sulfur and cadmium are distributed uniformly. The I–V curve and equivalent circuit of the metal–semiconductor–metal structure are shown in Fig. 1d, indicating that Schottky-type contacts are formed at both Cu-CdS and CdS-W interfaces[13,14].

Although no obvious shape changes are observed in the NW after biasing, as seen in Fig. 1g, the SAED pattern (Fig. 1h) shows two sets of diffraction spots from WZ CdS and low chalcocite (LC) Cu$_2$S. As mentioned in previous works, copper sulfide containing both sub-stoichiometric djurleite (Cu$_{1.93-1.97}$S) and stoichiometric LC (Cu$_2$S) phases can be formed from CdS[3,15]. It has been shown that djurleite transforms to LC by elimination of a copper vacancy with electron reduction or any other process[3,15]. In addition, copper surplus (LC formation) is thermodynamically favourable in this system[15,16]. In the present study, based on the SAED pattern (Fig. 1h), we conclude that the as-synthesized Cu$_{2-x}$S region in the NW is LC phase (see Supplementary Table 1 for more details). Comparing the EELS mapping data before and after reaction (Fig. 1e,f and j,k, respectively), we can find that the S anion sublattice remains intact, while the Cd cation sublattice transforms into a core-shell structure with a Cd core and a Cu shell. The representative EELS spectra corresponding to elemental mapping TEM image are presented in Supplementary Fig. 3. As SAED confirmed that WZ CdS and LC Cu$_2$S were the only phases present (Fig. 1h), the product was identified to be a NW with a CdS/Cu$_2$S core-shell structure. Thus, the Cu-CdS-W structure is transformed to the Cu-Cu$_2$S/CdS-W structure. Figure 1i shows the I–V curve and the equivalent circuit for this device. It is clear that an Ohmic contact is formed at the Cu-Cu$_2$S interface, which

is in accordance with previous reports[17]. If the Schottky barrier of CdS-W interface is too high, the Cu-Cu$_2$S/CdS-W system would not be conductive at a positive voltage. This suggests that the Schottky barrier reduces at the end of the reaction, on which more will be discussed later in this study. In general, CdS is an n-type semiconductor due to sulfur deficiency, whereas Cu$_2$S is a p-type semiconductor due to the presence of copper vacancies. Therefore, we conclude that the observed unidirectional rectifying behaviour originates from both Cu$_2$S/CdS interface and CdS/W interface.

To investigate the CE from CdS to Cu$_2$S at the atomic scale, we took high-resolution TEM images. As shown in Fig. 2a, the CdS (purple colour) and Cu$_2$S (golden colour) are both crystalline with core-shell (CdS/Cu$_2$S) structure. A few stacking faults are observed in CdS (Fig. 2a) as highlighted by red arrows. They may be introduced in the process of the preparation of CdS NWs, such as the stacking faults in Supplementary Fig. 2c. Figure 2b shows the CdS/Cu$_2$S interface wherein the reaction zone[1] can be clearly identified. The reaction zone (area enclosed by the yellow dotted lines) spans several atomic layers (about 1 nm) and it was critical for the solid-phase CE reaction[1]. For nanocrystals, the width of reaction zone determines the morphological stability of reaction precursors. Within reaction zone, the crystal is in a structurally non-equilibrium state where both the cations and anions are mobile[1]. If the reaction zone spans the whole width of the crystal, the morphology of crystal prefers the thermodynamic more stable shape (sphere-like) before all the ions reach the final equilibrium positions. The LC Cu$_2$S (200) lattice fringes begin to appear along the NW shell instead of the WZ -CdS (001) lattice fringes, as verified from the fast Fourier transformation (FFT) patterns shown in Fig. 2c,d. The schematic diagram of CdS-Cu$_2$S heterostructure shown in Fig. 2e corresponds to the structure in Fig. 2b. From these images, we can also identify the

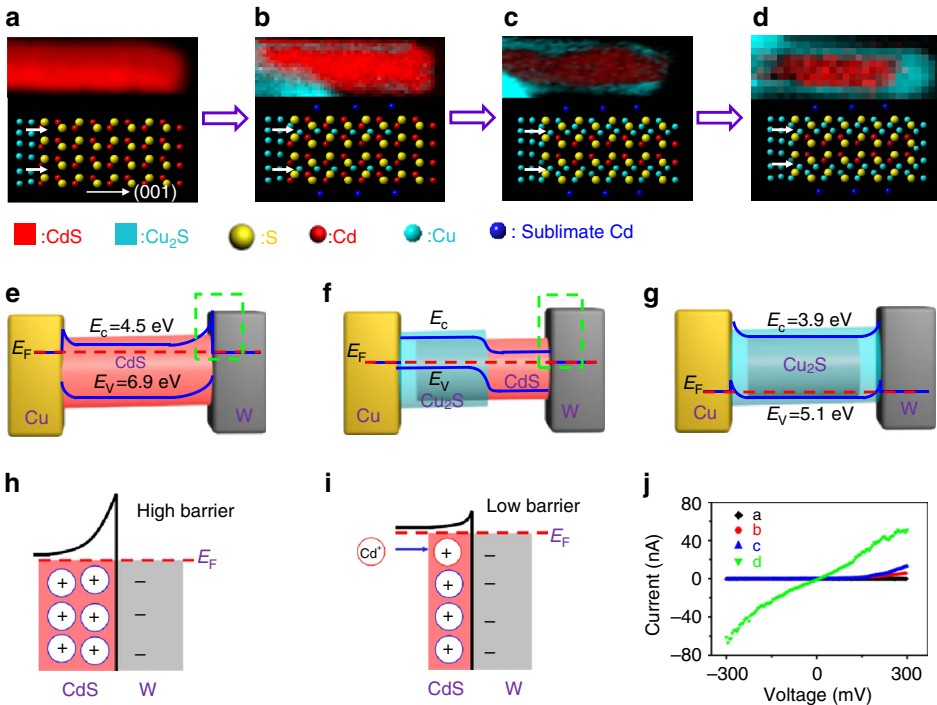

**Figure 3 | Formation process of CdS-Cu₂S core-shell structured NW.** (**a–d**) The top section displays STEM-EELS elemental (Cd and Cu) mapping images of the NW at different stages of the reaction and the bottom section illustrates the corresponding structures. Considering that the S sublattice in CdS NW is intact during the whole formation process, the Cd element distribution represents CdS and the Cu element represents Cu₂S in STEM-EELS mapping images. (**a**) The CdS NW, with (001) crystal orientation. (**b**) Cu cation migration along the subsurface of the NW in the early stage of the process. (**c**) Copper ions reach the other end of the NW, forming CdS/Cu₂S heterostructure with one open end. (**d**) Copper ions continue migrating, forming CdS/Cu₂S core-shell structure. (**e–g**) Energy band diagrams of Cu-CdS-W, Cu-Cu₂S-CdS-W and Cu-Cu₂S-W structures, corresponding to **a,c,d**, respectively. (**h**) Schematic illustration of the formation of a Schottky-type contact at the CdS/W interface in the green dotted line region of **e**. (**i**) Schematic illustration of the Schottky barrier reduced at the CdS/W interface in the green dotted line region of **f**. (**j**) I–V curves of the systems at different stages of reaction, corresponding to **a–d**.

orientation relationship of CdS$\{\bar{1}00\}$/$\{\bar{1}02\}$Cu₂S, which is in consistence with the previous observations on similar structures reported in the literature[18].

Using this synthesis technique, it is possible not only to grow CdS/Cu₂S core-shell structure with one end closed, but also to fabricate core-shell structures with both ends closed, as shown in Supplementary Fig. 4a–d. In addition, the degree of heterogeneity can be controlled by adjusting the biasing time. We applied a 0.5 V positive voltage on another NW and the EELS mapping data were obtained every 10 s. Results (Supplementary Fig. 4b–d) show that during the whole biasing process, the sulfide anion sublattice remains intact, and that the Cu₂S zone at the top end of the NW increases with biasing time. Supplementary Fig. 4e shows the lengths of the Cu₂S zones grown at the top end of the NW after biasing for different time. The growth rate of Cu₂S (marked by white two-way arrows) slows down with time, because it is increasingly harder for Cd ions to be replaced in the NW as the CE reaction progresses. Further biasing of partially cation-exchanged NWs will lead to deformation of the NWs before the total replacement of Cd ions (Supplementary Fig. 5a,b and Supplementary Movie 1) due to the high temperature raised by Ohmic heating.

**The physical mechanism of electrically driven CE.** The formation of CdS/Cu₂S core-shell structured NW was also studied using STEM–EELS mapping, where Cd maps represents CdS and Cu maps represents Cu₂S, as the S sub-lattice in the NW is intact during the whole exchange process. Before biasing (Fig. 3a), the starting template CdS NW is oriented in the <001> growth direction. During biasing, the following changes

occur. First, sublimation of Cd resulting in cation vacancies at the CdS surface due to Ohmic heating, while the inner Cd ions diffuse to the surface leaving cation vacancies at the subsurface. Second, electro-dissolution of the Cu electrode in accordance with the equation, $Cu \rightarrow Cu^+ + e^-$, after which the $Cu^+$ ions migrate along the subsurface of the NW towards the W electrode driven by the external electric field[8,9] to react with the sulfide ions according to the equation, $2Cu^+ + S^{2-} \rightarrow Cu_2S$ (Fig. 3b). Next, $Cu^+$ ions reach the other end of the NW forming CdS/Cu₂S core-shell structure with one end closed (Fig. 3c). Finally, $Cu^+$ continues to migrate forming CdS/Cu₂S core-shell structure with both ends closed (Fig. 3d). With continued biasing (Supplementary Fig. 4), the inward diffusion of $Cu^+$ will push $Cd^{2+}$ out of the cation locations and the expelled $Cd^{2+}$ will tend to escape through the weakest sites in the shell (marked by a blue arrow in Supplementary Fig. 4d).

Cd sublimation of heterogeneous NCs was previously observed in colloidal CdSe, CdSe/Cu₃P/CdSe NCs, NCs and CdSe/PbSe NCs[6,8–10]. The sublimation temperatures are as low as 450 K (ref. 8). When a constant bias is applied, the CdS NW's temperature will rise due to Ohmic heating until it reaches to a thermal equilibrium state. We simulated the temperature distribution of a CdS NW in thermal equilibrium state by finite element method and found that the temperature raised by Joule heating is enough to induce sublimation of Cd (Supplementary Fig. 6).

In this process, high chalcocite (HC) Cu₂S, which is the high-temperature phase of stoichiometric Cu₂S, is not only a reaction product but also functions as a channel for cation transport. This is due to the fact that there is a greater number of possible occupation sites with similar energy than the number of Cu atoms in the HC

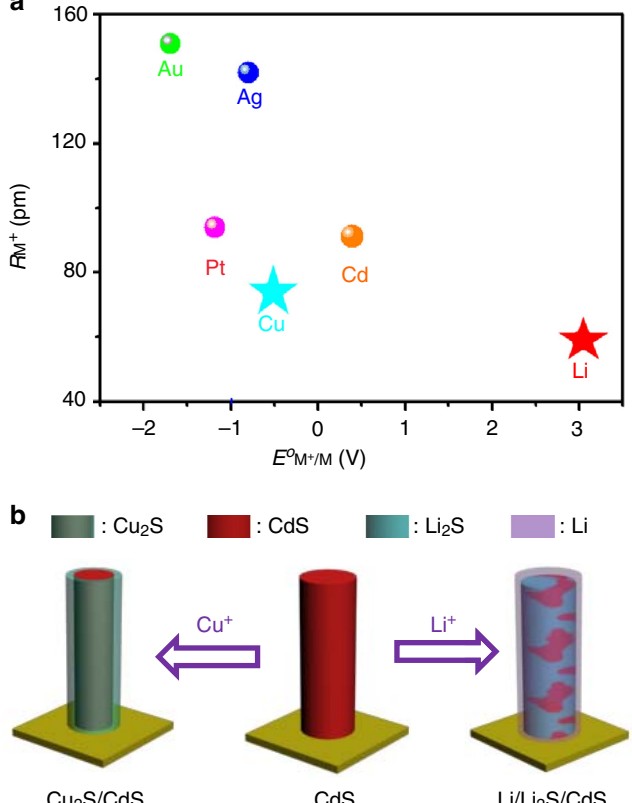

**Figure 4 | Applicability of the electrically driven CE method for other metals.** (**a**) Metal ion radius $R$ ($M^+$) versus standard electrode potential $E^o$($M^+/M$) diagram for different metals. The metals marked by spheres cannot exchange with CdS and those marked by stars can replace $Cd^{2+}$ in the NW (the values for standard electrode potential, Gibbs energies of formation and ionic radius of metals are shown in Supplementary Table 3). (**b**) Schematic diagram showing the formation of different heterostuctures after the reaction of CdS with $Cu^+$ and $Li^+$ for comparison.

phase[19]. The total reaction can be expressed by the equation:

$$2Cu^+ + CdS + 2e^- \rightarrow Cd \uparrow + Cu_2S \qquad (1)$$

In view of its much lower boiling point (765 °C) than copper (2561.5 °C)[20], all other conditions being the same, Cd is easier to sublimate. It should be emphasized that in the early stages of this process, the migration of $Cu^+$ is along the sub-surface rather than on the surface of the NW. Normally, $Cu^+$ is expected to move preferentially along the surface as compared with the bulk due to the smaller barrier[12]. However, the electro-dissolution of Cu is slower than the evaporation of Cd. As a result of Ohmic heating, Cd on the surface evaporates first following which the bulk $Cd^{2+}$ will migrate to the surface leaving behind cation vacancies in the subsurface. Under these conditions, it is easier for $Cu^+$ to migrate along the subsurface.

From Fig. 3 and Supplementary Fig. 4, we find that the reaction dominantly starts along [100] direction and then proceeds along [001] direction. According to the above analysis, the reaction is carried out on the premise of Cd sublimation. The Cu/CdS contact position has a higher temperature than other positions (Supplementary Fig. 6), so evaporation of Cd (resulting in Cd vacancies) starts at Cu/CdS interface first. It has been found that the interface CdS(001)/Cu₂S(001) has a lower formation energy than the attachment of $Cu_2S$ to the CdS (100) facet[18]. Thus, the reaction prefers to start along the [100] direction. Once Cd vacancies formed at the surface of the CdS NW, the ingoing Cu

ions tend to migrate more along the [001] direction under the action of electric field force. As a result, the reaction proceeds along [001] direction later.

The I–V curves of NWs at different stages of the reaction (Fig. 3a–d) are shown in Fig. 3j. The curve in Fig. 3a indicates that Schottky-type contacts are formed at both Cu-CdS and CdS-W interfaces[13,14] in accordance with the energy band diagram of the original Cu-CdS-W structure (Fig. 3e). The unidirectional rectifying behaviour of these structures shown in Fig. 3b,c can be explained on the basis of energy band diagram of Cu-Cu₂S/CdS-W system (Fig. 3f) and the origin of the bidirectional conductance of the structure in Fig. 3d can be traced back to the energy band diagram of the Cu-Cu₂S-W system[20,21] (Fig. 3g). The electrical properties of the NW depend on the extent of CE and can therefore be tuned to meet the needs of special functional devices. It is worth noting that the Schottky barrier of the CdS/W interface in Fig. 3e reduces upon biasing (Fig. 3f). Though the work function of W (4.55 eV)[20] is lower than the Fermi level of n-type CdS (about 5.7 eV)[20], it is still a Schottky contact at the CdS/W interface at the beginning (Fig. 3h) because of the Fermi level pinning effect[21]. The diffusion of $Cu^+$ will force the inclusion of a fraction of the $Cd^{2+}$ ions into the crystal lattice of the n-type CdS semiconductor, leading to heavy cation doping as shown in Fig. 3i (ref. 22). Positively charged Cd ions can provide an additional potential (Fig. 3i), thereby reducing both the built-in potential and the depletion width to maintain the Fermi energy balance between W and CdS[22]. Therefore, the Schottky barrier at the CdS/W interface reduces.

## Discussion

To confirm that this type of CE is electrically driven (including an electric field force and Ohmic heating), an equal negative voltage in the reverse direction (the direction of the electric field is from W tip to the Cu grid) was applied on the CdS NW to simulate a temperature distribution that is similar to the bias condition from the Cu grid to the W tip. The result of this simulation revealed that the CdS NW was not transformed into Cu₂S/CdS heterostructure under these conditions. This result indicates that the electric field force is the predominant driving force for CE. Nevertheless, the sublimation of cadmium and the electro-dissolution of the Cu electrode are actuated by Ohmic heating and, therefore, the kind of CE observed in our study is driven not only by electric field force but also by local Ohmic heating[4].

It is important to emphasize that the process observed in this study is different from other related electrically driven processes published hitherto (shown in Supplementary Table 2). The process of electromigration in a metal is driven by electronic wind force[23] and is not a chemical reaction. As indicated by previous publications[24], the electric field plays a role only in transferring lithium atoms to nanocrystals, and that the chemical reaction is driven by the strong reducibility of Li ions[24]. Anode materials in lithium or sodium ion batteries can be classified into three main groups depending on reaction mechanism[24] as shown in Supplementary Table 2: intercalation/de-intercalation materials, alloy/de-alloy materials and conversion materials. In both intercalation/de-intercalation and alloy/de-alloy materials, lithiation of the anode is through a combination reaction[24]. On the contrary, the electrically driven CE is a replacement reaction similar to the lithiation of a conversion materials anode. However, detailed analysis (Supplementary Fig. 7) shows that during lithiation of the anode conversion materials, the outgoing cations do not disappear and the anion sublattice suffers severe modification, which leads to significant changes in the size and shape. Therefore, the electrically driven CE is a new process that is unrelated to the lithiation of anode materials.

We tried to replace copper by several other metals (for example, gold, silver, platinum, aluminum, molybdenum, nickel and lithium) to produce similar heterostructures of NWs using this method. The results are shown in Fig. 4a. The metals marked by spheres denote those that cannot react with CdS and those marked by stars are metals that can replace $Cd^{2+}$ in CdS NWs. After an analysis of the results, we find that the following necessary conditions need to be fulfilled for the formation of heterostructures. First, the initial phase (hexagonal CdS) and the final phase (for example, $Au_2S$, $Ag_2S$, $Cu_2S$ and $Li_2S$) should have similar S sublattices to allow for a kinetically facile transformation to the new phase[4]. Second, these metals should have a low standard electrode potential so as to act as sources for the inserted cation[12]. For an individual NW, electrically driven CE reaction has a critical electrochemical potential as shown in Supplementary Fig. 8. Some constant voltages with variable amplitude (0–0.4 V, the interval is 0.1 V) were successively applied on a NW for 60 S, respectively. After each biasing, the SAED patterns were recorded (Supplementary Fig. 8a–e). We find SAED patterns of the NW would not change until the voltage was applied over the critical value (between 0.3 and 0.4 V), as displayed in Supplementary Fig. 8e. The corresponding EELS mappings of the NW (Supplementary Fig. 8g) reveal the formation of CdS/$Cu_2S$ heterostructure. In electrically driven CE reaction, Ag, Cu and Li are suitable electrochemically active electrodes as their standard electrode potentials (Supplementary Table 3), $E^o(Ag^+/Ag) = 0.799\,V$, $E^o(Cu^+/Cu) = 0.339\,V$ and $E^o(Li^+/Li) = -0.257\,V$, are not too large when compared with Au with $E^o(Au^+/Au) = 1.498\,V$ and Pt with $E^o(Pt^+/Pt) = 1.180\,V$, which means that they can be easily dissolved electrochemically[12]. It should be noted that Li is easily oxidized in air, but this does not affect our experimental results, as Li ions can migrate through the $Li_2O$ layer and take part in the displacement reaction[25]. As $Ag_2S$ is as good an ionic conductor as $Cu_2S$, the reaction may undergo similar to $Cu_2S$ formation. However, the CE reaction for Ag electrode was not observed. This might be owing to the deficient electrochemical potential supply to ionize silver before CdS NW melt.

The experimental results using Li as the electrochemically active electrode are significantly different. Supplementary Fig. 9a shows the single crystal CdS NW with the WZ structure. After biasing, it is transformed into a polycrystalline $Li_2S$/CdS heterostructure with a thin amorphous Li shell and the radius is increased (Supplementary Fig. 9b). High-resolution TEM images shown in Supplementary Fig. 9c,d are those of one CdS crystal grain and one $Li_2S$ crystal grain in the polycrystalline $Li_2S$/CdS heterostructure, respectively. From the EELS mapping data, we found that the S anion sublattice had little deformation and the anion sublattice size remained almost unchanged, which is similar to the situation for Cu. The increase in diameter is therefore attributed to the presence of a shell of Li on the outer surface, as shown in Supplementary Fig. 9e.

The reaction dynamics and the transport pathway are also different for the reaction of lithium with CdS as compared with that of copper with CdS. For copper, a $Cu_2S$/CdS core-shell structure was formed after biasing, whereas for lithium the reaction resulted in a mixed heterostructure of $Li_2S$ and CdS with a thin amorphous Li shell (Fig. 4b). This difference is due to differences in both ion migration rates and in the ease of bonding between the migrating cations and the sulfur sublattice. The displacement reaction of CdS with copper is actuated not only by the electric field force but also by Ohmic heating. The elimination of cadmium by evaporation is caused by the decomposition of CdS by Ohmic heat energy and the inward migration of $Cu^+$ is driven by electric field force. As the standard Gibbs free energy of formation of $Cu_2S$[20] ($\triangle G_f^o(Cu_2S) = -86.2\,KJ\,mol^{-1}$) is much

lower than that of CdS[20] ($\triangle G_f^o(CdS) = -156.5\,KJ\,mol^{-1}$), the interaction of the $Cu^+$ with the $S^{2-}$ is weaker than that of the $Cd^{2+}$. In the absence of Ohmic heating, $Cu^+$ may prefers to migrate through the NW rather than diffuses inward to displace the $Cd^{2+}$. Furthermore, as the reaction product (HC phase $Cu_2S$) is a super ionic conductor, this would be as a faster transport path to supply Cu ions for subsequent reactions along the axial direction. Finally, a $Cu_2S$/CdS core-shell structure is formed. For Li ion exchange, the driving forces include chemical affinity in addition to electric field force and Ohmic heating, because Li ions show a much stronger interaction[20] ($\triangle G_f^o(Li_2S) = -225.0\,KJ\,mol^{-1}$) with sulfide anions when compared with Cd ions[20] ($\triangle G_f^o(CdS) = -156.5\,KJ\,mol^{-1}$). Hence, Li ions are more likely to diffuse inward and displace Cd ions. The reaction between Li and CdS may take place even without assistance from Ohmic heating. However, the ionic diffusivity of Li in $Li_2S$ is very poor[26] as a result of which most of the Li ions migrate along the NW surface as demonstrated by the presence of a thin shell of Li on the NW surface (Supplementary Fig. 9e). This leads to the transformation of the NW into a mixed heterostructure at the end of reaction.

In conclusion, we demonstrate an in situ CE process driven by electricity to transform individual NW into heterostructures during TEM imaging. This method can be used not only to finely control the CE process but also to selectively modify individual nanocrystals in an integrated system of NCs. The various reaction steps in the process are identified by observing the growth of CdS/$Cu_2S$ core-shell structure from the CdS NW precursor, which are namely sublimation of Cd, electro-dissolution of Cu, migration of Cu ions and the formation of CdS/$Cu_2S$ core-shell structure. The reaction mechanism strongly supports the fact that the inwardly migrating cations (here, $Cu^+$ and $Li^+$) originate from metal active electrodes and are driven by the electric field force, whereas the outward migration of cations (here, $Cd^{2+}$) occurs due to evaporation into the gas phase and is actuated by Ohmic heating. Results from contrasting experiments reveal that the structures of the heterogeneous nanocrystals formed depend on both the mobility of the cations within the crystal and the binding energy of the inwardly migrating ions and the sulfur sublattice. Our results open a new perspective to selective CE on individual nanocrystals and also provide critical insights into the microscopic mechanism of solid-state exchange processes.

## Methods

**CdS NW preparation.** CdS NW samples were synthesized via a solvothermal method. In brief, 0.2665 g $Cd(CH_3COO)_2\cdot 2H_2O$ (99.99%, Aladdin) and 0.0641 g sublimed sulfur (99.95%, Aladdin) were dissolved in 40 ml ethylenediamine (98%, Aladdin) under vigorous stirring and then transferred to a teflon-lined stainless steel autoclave. The autoclave was sealed and heated at 200 °C for 2 h after which it was allowed to cool to room temperature naturally. The yellowish products were isolated by centrifugation at 6,000 r.p.m. for 10 min and washed five times with methanol (certified ACS, Aladdin).

**Set up of the in situ TEM experiment.** In this work, a half copper grid was glued onto a gold wire with conductive epoxy. The as-prepared CdS NWs were dissolved in ethanol solution to achieve a homogeneous suspension. Then, a drop of solution was casted on the half Cu grid and was dried in the air for 10 min. As a result, some CdS NWs were anchored on the edge of the half grid. The tungsten tips were prepared by electrochemically etching. The etching solution was 2 mol l$^{-1}$ NaOH and the etching voltage was 2 V with a compliant current of 10 mA. All W tips were cleaned by plasma cleaning for 60 S to reduce impurities at the surface for in situ TEM experiments. The structure of the device is shown in Fig. 1a. All the in situ experiments in this work were conducted in vacuum.

**Materials characterization and electrical measurements.** Powder X-ray diffractograms of the obtained CdS products were measured using an X-ray diffractometer (Rigaku, Ultima III) with Cu Kα radiation as the X-ray source. The

TEM characterization and electrical property measurements were carried out using an aberration-corrected TEM (Titan 80–300) equipped with a TEM–scanning tunneling microscopic holder.

**Simulation of temperature gradient.** We supposed that the CdS NW as a one-dimensional system with length $L$ obeys the classical heat equation with Joule heating:

$$\nabla(k\nabla T) + \frac{\delta V^2}{L^2} = 0 \qquad (2)$$

where $V$ is the applied bias, $L$ is the length of NW, $k$ is the thermal conductivity, $\delta$ is the electric conductivity and $T$ is the temperature. The parameters of $k$ and $\delta$ of Cu[20,27], CdS[28–30] and W[20,27] are well-known. Equation (2) was solved by finite element method with appropriate boundary conditions at a given applied bias of 1 V.

**Data availability.** The Cu and W thermal conductivity data that support the findings of this study are available in Lange's Handbook of Chemistry, http://fptl.ru/biblioteka/spravo4niki/dean.pdf. The Cu and W electric conductivity data that support the findings of this study are available in *J. Phys. Chem. Ref.* data, https://nist.gov/JPCRD/jpcrdS1Vol3.pdf The CdS thermal conductivity data that support the findings of this study are available in *J. Appl. Phys.* with the identifier doi: 10.1063/1.3476469 (ref. 30). The CdS electric conductivity data that support the findings of this study are available in *Appl. Phys. Lett.* with the identifier doi: 10.1063/1.1900950 (ref. 29).

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

## Acknowledgements

This work was supported by the National Natural Science Foundation of China (numbers 51420105003, 61274114, 11327901, 11525415, 11674047, 11604047 and 11504046) and the Fundamental Research Funds for the Central Universities (numbers 2242016K41039, MTEC-2015M03, 2242015KD014 and NJ20150026). H.Z. thanks the support of DOE BES Materials Sciences and Engineering Division Under Contract No KC22ZH.

## Author contributions

L.S. and H.Z. conceived the project. Q.Z. performed the *in situ* TEM imaging. Q.Z. and K.Y. carried out the data analysis. Q.Z., K.Y., H.Z. and L.S. co-wrote the paper with all authors contributing to the discussion and preparation of the manuscript.

## Additional information

**Competing interests:** The authors declare no competing financial interests.

