## [Peer Review File · Nature Communications]

Reviewers' comments:

Reviewer #1 (Remarks to the Author):

This an overall interesting paper and in my view it could deserve publication, after the following points have been clarified:

- 1) The authors claim that the starting nanowires used for the in situ cation exchange are aligned vertically on a Cu substrate. However, no details are given about the deposition and the alignment steps and the actual morphology of the resulting film made of vertically aligned wires.
- 2) The process of the Electrically Driven Cation Exchange is well described. It is shown that after a certain time (30 sec at 0.5 V positive voltage) a complete Cu₂S shell has grown onto the starting CdS NW (Figure S3d). What happens if the NW is biased at positive voltage for longer time? Can the CE be driven to the total replacement of existing Cd²⁺ ions (as it happens in the thermally activated CE reaction) or is it a "self limiting" process?

Reviewer #2 (Remarks to the Author):

Review Comment on the Manuscript NCOMMS-16-23712-T:

Q. Zhang et al. reported the method to control cation exchange of CdS nanowires in a single nanowire level. The cation exchange reaction described here is very unique not only in their high precision reaction control, but also in a reaction pathway that the reaction starts from the surface, not from the stacking faults defects, which is different from previously reported cation exchange reactions.

Although the experimental observation described here has strong potential to be published in Nature Communications, the discussions about the observation is in primitive stage.

Most of all, in order to classify the nanocrystal reaction as cation exchange reaction rather than electrochemical conversion reaction, the applied potential should only be used for ionization of copper or lithium, not for the reduction of cadmium ion inside CdS lattice. The described reaction seems to be a combination of both electrochemical and cation exchange reaction. (i) The major reaction pathway is speculated to be electrochemical conversion reaction, but differs from the previous reports in terms of the product, which almost maintains its morphology. (ii) But it would be cation exchange reaction, in which zero-valent cadmium vaporization would be the driving force for the spontaneous and

irreversible reaction. Then, the reaction would be electrochemical reaction analogy of previously reported ion solvation-driven cation exchange reaction (Science 2004, 306, 1009-1012). I suggest to re-consider the manuscript for publication in Nature Communications after major revision.

1. "The aqueous solubility-driven cation exchange" is not a right term, because various cation exchange reaction occurs in non-aqueous reaction environment of methanol, phosphine, etc. It would be better to use the term "ion solvation-driven cation exchange reaction".
2. Is ref.19 the most relevant one for your discussion about the 1~2 nm solid-state reaction zone?
3. Detailed explanation on the electrochemical cell configuration is missing in supplementary information. Is the cell filled with electrolyte (liquid cell) or in vacuum?
4. Please provide representative EELS spectra corresponding to elemental mapping TEM image in supplementary information.
5. Although the discussion of IV curve in page page 11 is very convincing, but it's too speculative. It is difficult to figure out the actual origin of the rectification behavior. From figure 3, although the energy barrier may be reduced, the junction between n-CdS and W is still Schottky-type rather than Ohm-like. Therefore it is speculated that the rectification behavior is originated from both Cu₂S/CdS interface and CdS/W interface. In this regard, the discussion from sentence line 128 to 131 not that necessary, particularly the last sentence "the LC phase is more suitable for photovoltaic applications."
5. The discussion on the mechanism of the electrically driven cation exchange in page 10 and 11 is well-specified. However, it lacks the discussion on Joule heating which induces sublimation of Cd. Or could you provide general discussion on Joule heating in the in-situ TEM electrochemical reaction condition?
6. From figure 3 and figure S3, it is very interesting that the reaction starts along dominantly (100) direction and then proceeds along (001) direction. Why?
7. The authors claim that cation exchange of CdS to Ag₂S didn't occur because of larger radius of Ag⁺ than Cd²⁺ radius (92 pm). This is not true. The cation exchange reaction was first observed in the ion solvation-driven conversion reaction of CdS to Ag₂S (Science 2004, 306, 1009-1012). It would be

owing to the deficient electrochemical potential supply to ionize silver. As Ag_2S is good ionic conductor as Cu_2S , the reaction may undergo similar to Cu_2S formation when enough potential is applied.

8. Could you provide the applied electrochemical potential control experiments? This would be sophisticated and difficult experiments to control. However, it would be of importance for the reaction mechanism discussion as previously mentioned.

9. All the references should be followed by Nature guideline.

Reviewer #3 (Remarks to the Author):

Overall, I find this to be a fascinating work, worth publication in this journal. The ability to selectively exchange an individual nanocrystal is unprecedented. That said, the uniqueness of the TEM holder for this process does make it less useful to the broader community as a means for exchange as part of a large process. What makes this work special is its ability to elucidate the mechanism by which the cation exchange process proceeds, and understand what unique properties partially exchanged heterostructures may present. While I do like this work, I have several concerns about the arguments that are made, and which I have outlined below:

1. I am not convinced that there is a complete core/shell structure as is suggested throughout the article. It seems to me that there is exchange in areas, but it is not uniform, and not surrounding the entire wire like a shell.

a. Figure 1, why is there not Cu shown everywhere in the After EELS? If this were core/shell, then it should be on top and bottom, not just the sides.

b. Figure 2, shouldn't there be interfering diffraction patterns in the region on the right, where there should be copper sulfide on top of the cadmium sulfide? Since the crystal structures are epitaxial, both should be observed in the one region.

c. Figures 3f-g suggest a different mechanism from core/shell growth. These seem to say you have complete conversion moving down the length of the rod. I see how this does a better job explaining the electrical behavior, but it doesn't really match the 3a-d figures at all.

2. The observed stacking faults discussed in line 147, are attributed to strain during growth, but CdS frequently has stacking faults. I would need evidence that these did not exist before exchange to be convinced that they were formed during exchange by the strain.
3. In line 150, the authors mention that the reaction zone is critical for the solid-phase CE reaction. I would like to have this elaborated. What exactly is being suggested? To me this is simply the region between the two crystal phases, where neither one completely exists, and it's more of an amorphous region. But is there something more being suggested? Is there something special mechanistically occurring here?
4. The chemical reaction in line 194 troubles me. You have copper ions coming from the electrode, you have electrons being injected, but where is the elemental sulfur coming from? Shouldn't that already be S²⁻ from the CdS lattice? Or is it being oxidized at some point?
5. Line 324, the authors claim this allows them to control the synthesis of nanocrystals at the atom level, but this is no more true than the same claim for solution processed cation exchange. They have demonstrated selective cation exchange on individual nanocrystals, which is fantastic, but this is still not controllable at the atom level any more so that other cation exchange methods.
6. Looking at the Li exchange process, I agree that the process is more patchy and less amenable to the exchange internally, but I'm not convinced that the lithium goes into the structure at all. To me it looks like it's just sitting on the surface, especially in figure 5e.

Replies to the Referees' comments:

Reviewer # 1:

This an overall interesting paper and in my view it could deserve publication, after the following points have been clarified:

1.

The authors claim that the starting nanowires used for the in situ cation exchange are aligned vertically on a Cu substrate. However, no details are given about the deposition and the alignment steps and the actual

morphology of the resulting film made of vertically aligned wires.

Authors' Reply:

We thank the referee for the suggestions. The detail experimental information was added in the revised manuscript (Page 14, Line 302-310). In this work, a half copper grid was glued onto a gold wire with conductive epoxy. The as-prepared CdS nanowires were dissolved in ethanol solution to achieve a homogeneous suspension. Then, a drop of solution was casted on the half copper grid, and was dried in the air for 10 minutes. As a result, some CdS nanowires were anchored on the edge of the half copper grid. Then, we can selectively modify the anchored CdS nanowires through a TEM-STM platform. The vertically aligned CdS nanowires on Cu substrate presented in Figure (1 a) is just in order to vividly demonstrate the selective cation exchange on individual nanocrystals. (The Figure (1 a) is also modified to show the actual situation)

2. The process of the Electrically Driven Cation Exchange is well described. It is shown that after a certain time (30 sec at 0.5 V positive voltage) a complete Cu₂S shell has grown onto the starting CdS NW (Figure S3d). What happens if the NW is biased at positive voltage for longer time? Can the CE be driven to the total replacement of existing Cd²⁺ ions (as it happens in the thermally activated CE reaction) or is it a “self limiting” process?

Authors' Reply:

Thanks for the nice suggestion. Further biasing of partially cation-exchanged nanowires can lead to deformation of the NW before the total replacement of Cd ions (supporting information Figure S5 and movie S1). We simulated the temperature distribution of a CdS nanowire in thermal equilibrium state by finite element method (Fig. S6) and found that the temperature raised by Joule heating is too high. So, long-time biasing will lead to melting of NW. We have added a brief explanation in revised version (Page 6, Line 133-135).

Reviewer # 2:

Q. Zhang et al. reported the method to control cation exchange of CdS nanowires in a single nanowire level. The cation exchange reaction described here is very unique not only in their high precision reaction control, but also in a reaction pathway that the reaction starts from the surface, not from the stacking faults defects, which is different from previously reported cation exchange reactions. Although the experimental observation described here has strong potential to be published in Nature Communications, the

discussions about the observation is in primitive stage. Most of all, in order to classify the nanocrystal reaction as cation exchange reaction rather than electrochemical conversion reaction, the applied potential should only be used for ionization of copper or lithium, not for the reduction of cadmium ion inside CdS lattice. The described reaction seems to be a combination of both electrochemical and cation exchange reaction. (i) The major reaction pathway is speculated to be electrochemical conversion reaction, but differs from the previous reports in terms of the product, which almost maintains its morphology. (ii) But it would be cation exchange reaction, in which zero-valent cadmium vaporization would be the driving force for the spontaneous and irreversible reaction. Then, the reaction would be electrochemical reaction analogy of previously reported ion solvation-driven cation exchange reaction (Science 2004, 306, 1009-1012). I suggest to re-consider the manuscript for publication in Nature Communications after major revision.

1. “The aqueous solubility-driven cation exchange” is not a right term, because various cation exchange reaction occurs in non-aqueous reaction environment of methanol, phosphine, etc. It would be better to use the term “ion solvation-driven cation exchange reaction”.

Authors' Reply:

Thank the referee for the suggestion, we agree with you. We have made the corrections in revised manuscript (Page 1, Line 16-17; Page 2, Line 33).

2. Is ref.19 the most relevant one for your discussion about the 1~2 nm solid-state reaction zone?

Authors' Reply:

We have corrected the reference 19 in revised manuscript (Page 5, Line 114).

3. Detailed explanation on the electrochemical cell configuration is missing in supplementary information. Is the cell filled with electrolyte (liquid cell) or in vacuum?

Authors' Reply:

Thank you for the suggestion. We have added detail information about the cell configuration of the in situ TEM experiment in the revised manuscript (Page 14, Line 302-310). In this work, a half copper grid was glued onto a gold wire with conductive epoxy. The as-prepared CdS NWs were dissolved in ethanol solution to achieve a homogeneous suspension. Then, a drop of solution was casted on the half Cu grid, and was dried in the air for 10 minutes. As a result, some CdS NWs were anchored on the edge of the half grid. The tungsten tips were prepared by electrochemically etching. The etching solution was 2 mol L⁻¹ NaOH and the etching voltage was 2 V with a compliant current of 10 mA. All W tips were cleaned by plasma cleaning for 60 S to reduce impurities at the surface for in-situ TEM experiments. The structure of the device is shown in Fig. 1a. The cell was in vacuum and no electrolyte was needed.

4. Please provide representative EELS spectra corresponding to elemental mapping TEM image in supplementary information.

Authors' Reply:

Thank the referee for the suggestion. We have provided representative EELS spectra corresponding to the CdS nanowire precursor, the core of the Cu₂S/CdS core shell structure nanowire and the shell of the Cu₂S/CdS core shell structure nanowire in revised supplementary information, respectively (Supplementary information, Figure S3; Revised manuscript, Line 94-95).

5. Although the discussion of IV curve in page 11 is very convincing, but it's too speculative. It is difficult to figure out the actual origin of the

rectification behavior. From figure 3, although the energy barrier may be reduced, the junction between n-CdS and W is still Schottky-type rather than Ohm-like. Therefore it is speculated that the rectification behavior is originated from both Cu₂S/CdS interface and CdS/W interface. In this regard, the discussion from sentence line 128 to 131 not that necessary, particularly the last sentence “the LC phase is more suitable for photovoltaic applications.”

Authors' Reply:

Thanks for the nice comments. We reconsider the contact type of CdS/W interface, and agree with your suggestion. The corresponding explanations have been changed in the revised manuscript (Page 5, Line 99-101, 104-105; Page 9, Line 187-189, 195). We also deleted the discussion from sentence line 128 to 131 in old manuscript.

6. The discussion on the mechanism of the electrically driven cation exchange in page 10 and 11 is well-specified. However, it lacks the discussion on Joule heating which

induces sublimation of Cd. Or could you provide general discussion on Joule heating

in the in-situ TEM electrochemical reaction condition?

Authors' Reply:

Thanks for the nice suggestion. We added the discussion on Joule heating in the revised manuscript (Page 7, Line 151-156). The Cd sublimation was previously reported in colloidal CdSe NCs, CdSe/Cu₃P/CdSe NCs and CdSe/PbSe NCs. The sublimation temperature is even lower than 450 K (Nano Letter 14, 3661-3667 (2014)). When a constant bias is applied, the CdS NW's temperature will rise due to Ohmic heating until reaching a thermal equilibrium state. We simulated the temperature distribution of a CdS nanowire in thermal equilibrium state by finite element method (Fig. S6) and found that the temperature raised by Joule heating is enough to induce sublimation of Cd.

7. From figure 3 and figure S3, it is very interesting that the reaction starts along dominantly (100) direction and then proceeds along (001) direction. Why?

Authors' Reply:

Thanks for the nice suggestion. We have added the explanation about this phenomenon in revised manuscript (Page 8, Line 171-179). According to the analysis of electrically driven cation exchange, the reaction is carried out on the premise of Cd sublimation. The Cu/CdS contact position has a higher temperature than other positions, so evaporation of Cd (resulting in Cd vacancies) starts at Cu/CdS interface first.

It has been found that the interface CdS(001)/Cu₂S(001) has a lower formation energy than the attachment of Cu₂S to the CdS (100) facet (J. Am. Chem. Soc.131, 5285-5293

(2009)). Thus, the reaction prefers to start along [100] direction. Once Cd vacancies formed at the surface of the CdS NW, the ingoing Cu ions tend to migrate along the [001] direction under the action of electric field force. As a result, the reaction proceeds along [001] direction later.

8. The authors claim that cation exchange of CdS to Ag₂S didn't occur because of larger radius of Ag⁺ than Cd²⁺ radius (92 pm). This is not true. The cation exchange reaction was first observed in the ion solvation-driven conversion reaction of CdS to Ag₂S (Science 2004, 306, 1009-1012). It would be owing to the deficient electrochemical potential supply to ionize silver. As Ag₂S is good ionic conductor as Cu₂S, the reaction may undergo similar to Cu₂S formation when enough potential is applied.

Authors' Reply:

Thanks for the nice suggestion. We agree with your explanation. We have corrected the explanation in revised manuscript (Page 11, Line 242-245). As Ag₂S is a good ionic conductor as Cu₂S, the reaction may undergo

similar to Cu_2S formation. However, the CE reaction for Ag electrode was not observed. This might be owing to the deficient electrochemical potential supply to ionize silver before CdS NW melt.

9. Could you provide the applied electrochemical potential control experiments? This would be sophisticated and difficult experiments to control. However, it would be of importance for the reaction mechanism discussion as previously mentioned.

Authors' Reply:

Thanks for the nice suggestion. We provided a qualitative control experiment with different applied voltages (Supporting information Figure S8) and added some explanations in revised manuscript (Page 11, Line 230-236). For an individual nanowire, the electrically driven cation exchange reaction indeed has a critical electrochemical potential. But, for different nanowires, they may have different critical electrochemical potentials as the length and the contact resistances of nanowires might be different.

10. All the references should be followed by Nature guideline.

Authors' Reply:

Thanks for your suggestion. We have corrected the format of references followed by Nature guideline.

Reviewer # 3:

Overall, I find this to be a fascinating work, worth publication in this journal. The ability to selectively exchange an individual nanocrystal is unprecedented. That said, the uniqueness of the TEM holder for this process does make it less useful to the broader community as a means for exchange as part of a large process. What makes this work special is its ability to elucidate the mechanism by which the cation exchange process proceeds, and understand what unique properties partially exchanged heterostructures may present. While I do like this work, I have several concerns about the arguments that are made, and which I have outlined below:

1. I am not convinced that there is a complete core/shell structure as is suggested

throughout the article. It seems to me that there is exchange in areas, but it is not uniform, and not surrounding the entire wire like a shell. a. Figure 1, why is there not Cu shown everywhere in the After EELS? If this were core/shell, then it should be on top and bottom, not just the sides.

b. Figure 2, shouldn't there be interfering diffraction patterns in the region on the right, where there should be copper sulfide on top of the cadmium sulfide? Since the crystal structures are epitaxial, both should be observed in the one region. c. Figures 3f-g suggest a different mechanism from core/shell growth. These seem to say you have complete conversion moving down the length of the rod. I see how this does a better job explaining the electrical behavior, but it doesn't really match the 3a-d figures at all.

Authors' Reply:

a. We thank the referee for the comments. It is a complete core/shell structure after reaction. The EELS spectrum from the core of CdS/Cu₂S (Figure S3b) shows the signal of Cu, which demonstrates the presence of Cu. The signal of Cu in the core part is weaker than that in the side part as more electrons will be scattered due to the existence of CdS. In a certain contrast, the weaker Cu signal can be invisible. We are sorry for the using of unsuitable contrast and causing the puzzle. We have modified the image in the revised manuscript (Figure 1k). The separate Cu element mappings are shown in Figures R1.

Figure R1| STEM EELS mappings of Cu element. a, Corresponding to the NW in Figure 1K. b-d, Corresponding to the NW in Figure S4b-d, respectively.

b. We thank the referee for the comments. As mentioned above, an unsuitable contrast can make the weak signal invisible. We changed the contrast of FFT by Gatan Digital micrograph software and indeed found some diffraction points belonging to Cu_2S (the revised Figure 2d). The enlarged FFT is shown below (Figures R2), and the diffraction points belonging to Cu_2S are pointed by yellow arrows.

Figure R2| The enlarged picture of Figure 2d. Yellow arrows point out the diffraction points belonging to Cu₂S.

c. Figure 3e-f are the structure models and the band structure diagrams corresponding to Cu-CdS-W, Cu-Cu₂S-CdS-W and Cu-Cu₂S-W structures, respectively. For Figure 3b and Figure 3c, Cu ions migrate into CdS bulk forming Cu₂S, then Cu-Cu₂S contact was built at one end. In the other end, the contact was still between CdS and W, therefore we use the Cu-Cu₂S-CdS-W model to describe Figure 3b and Figure 3c. For Figure 3d, the completely closed shell was formed, so the two electrodes are both connected to Cu₂S, which corresponds to the Cu-Cu₂S-W model. The electrical behaviors shown in Figure 3j are also consistent with the models. In order to make the models more suitable, we changed the transparency of them (Figure 3e-g) in the revised manuscript.

2. The observed stacking faults discussed in line 147, are attributed to strain during growth, but CdS frequently has stacking faults. I would need evidence that these did not exist before exchange to be convinced that they were formed during exchange by the strain.

Authors' Reply:

We thank the referee for the comments, we agree that the stacking

faults are already exist before reaction. We carefully checked a mass of TEM images of the as-prepared CdS nanowires, and confirmed CdS nanowires indeed have stacking faults, as shown in Fig. R3. We have corrected the explanation of stacking faults in the corresponding section of

the manuscript (Page 5, Line110-111).

Figure R3. Stacking faults (red arrows pointed) are presented in CdS nanowires (a) and in the initial CdS nanowire precursor (b).

3. In line 150, the authors mention that the reaction zone is critical for the solid-phase CE reaction. I would like to have this elaborated. What exactly is being suggested? To me this is simply the region between the two crystal phases, where neither one completely exists, and it's more of an amorphous region. But is there something more being suggested? Is there something special mechanistically occurring here?

Authors' Reply:

Thanks for the nice suggestion. We agree with that the reaction zone is an amorphous region (Annu. Rev. Mater. Res. 33, 55, 2003). We suggested that the width of reaction zone determines the morphological stability of reaction precursors. Within reaction zone, the crystal is in a structurally non-equilibrium state where both the cations and anions are mobile. If the reaction zone spans the whole width of the crystal, the morphology of crystal prefers to be sphere

instead of keeping the original morphology (Science 306, 1009-1012 (2004)). We added the explanation in the revised manuscript (Page 5-6, Line114-118).

4. The chemical reaction in line 194 troubles me. You have copper ions coming from the electrode, you have electrons being injected, but where is the elemental sulfur coming from? Shouldn't that already be S²⁻ from the CdS lattice? Or is it being oxidized at some point?

Authors' Reply:

We thank the referee for the comments. The chemical reaction “ $2\text{Cu}^+ + \text{S} + 2\text{e}^- \rightarrow \text{Cu}_2\text{S}$ ” is changed to “ $2\text{Cu}^+ + \text{S}^{2-} \rightarrow \text{Cu}_2\text{S}$ ”. We have changed the equation in the revised manuscript (Page 7, Line146).

5. Line 324, the authors claim this allows them to control the synthesis of nanocrystals at the atom level, but this is no more true than the same claim for solution processed cation exchange. They have demonstrated selective cation exchange on individual nanocrystals, which is fantastic, but this is still not controllable at the atom level any more so than other cation exchange methods.

Authors' Reply:

Thanks for the nice suggestion. We agree your opinion. We have changed the statement in the revised manuscript (Page 13, Line 289).

6. Looking at the Li exchange process, I agree that the process is more patchy and less amenable to the exchange internally, but I'm not convinced that the lithium goes into the structure at all. To me it looks like it's just sitting on the surface, especially in figure

5e.

Authors' Reply:

Thanks for the nice suggestion. As displayed in Figure S9e in revised supporting information, the outmost-shell only have Li element. So it

looks like lithium is just sitting on the surface. However, it should be noted that the HRTEM images of the NW demonstrates the presence of Li_2S (Figure S9d). In the whole process, S anion sublattice almost unchanged, the Li ions diffuse on the surface of CdS nanowire. So there must be some Li ions go into the structure to replace Cd ions forming patchy Li_2S . The EELS mapping of Cd also confirms that some Cd ions were replaced. The STEM-EELS mappings of Cd, Li and S elements are shown in Figure R4. The Schematic diagram (Figure 4b) have been changed in the revised manuscript.

Figure R4| STEM-EELS mappings of Cd, Li and S for the nanowire in Figure S9 b. From them we can see that the outmost shell only has Li element. Meanwhile, some Cd ions are replaced by Li ions.

Summary of the changes:

(Changes in the revised manuscript are highlighted in red color)

1. In Page 1, “College of Materials Science and Engineering, Nanjing Tech University, Nanjing 210096, P. R. China.” is added in line 13.
2. In Page 1, Line 16-17, “aqueous solubility activated CE” is changed to “ion solvation- driven CE reaction”, and “reaction” is added after “thermally activated CE”.

3. In Page 1, Line 23, “this” is changed to “a”, and “controlled” is changed to “selective”.
4. In Page 2, Line 33, “aqueous solubility” is changed to “ion solvation”.
5. In Page 3, Line 55-56, “Results” and “Construction of the experimental setup” are added.
6. In Page 4, Line 73, “Electrically driven cation exchange” is added.
7. In Page 4-5, Line 94-95, “The representative EELS spectra corresponding to elemental mapping TEM image are presented in Fig. S3.” is added.
8. In Page 5, Line 99-100, “If a Schottky contact was formed at the CdS-W interface” is changed to “If the Schottky barrier of CdS-W interface is too high”.
9. In Page 5, Line 101, “collapses” is changed to “reduces”.
10. In Page 5, Line 104-105, “Therefore, we conclude that the CdS/Cu₂S (p-n) junction is responsible for the observed unidirectional rectifying behavior observed in the I-V curve.” is changed to “Therefore, we conclude that the observed unidirectional rectifying behavior originates from both Cu₂S/CdS interface and CdS/W interface”.
11. In Page 5, Line 110-111, “likely due to strain force” is changed to “They may be introduced in the process of the preparation of CdS NWs, like the stacking faults in Fig.

S2c”.

12. In Page 5, Line 114, reference “19” is changed to reference “1”.

13. In Page 5-6, Line 114-118, “For nanocrystals, the width of reaction zone determines the morphological stability of reaction precursors. Within reaction zone, the crystal is in a structurally non-equilibrium state where both the cations and anions are mobile¹. If the reaction zone spans the whole width of the crystal, the morphology of crystal prefers the thermodynamic more stable shape (sphere-like) before all the ions reach the final equilibrium positions.” is added after “The reaction zone (area enclosed by the yellow dotted lines) spans several atomic layers (about 1 nm) and it was critical for the solid-phase CE reaction¹.”.
14. In Page 6, Line 133-135, “Further biasing of partially cation-exchanged nanowires will lead to deformation of the NWs before the total replacement of Cd ions (Fig. S5a-c and Movie S1) due to the high temperature raised by Ohmic heating.” is added.
15. In Page 6, Line 137, “The physical mechanism of electrically driven cation exchange.” is added.
16. In Page 7, Line 146, “ $2\text{Cu}^+ + \text{S} + 2\text{e}^- \rightarrow \text{Cu}_2\text{S}$ ” is changed to “ $2\text{Cu}^+ + \text{S}^{2-} \rightarrow \text{Cu}_2\text{S}$ ”.
17. In Page 7, Line 151-156, we have added some discussions about the influence of Joule heating on sublimation of Cd.
18. In Page 8, Line 171-179, we have added some explanations about the cation exchange reaction starts along dominantly (100) direction and then proceeds along (001) direction.
19. In Page 9, Line 187-189, “It’s worth noting that the Schottky-type contact of the CdS/W

interface in Fig. 3e turns into an Ohmic contact upon biasing (Fig. 3f).” is changed to

“It is worth noting that the Schottky barrier of the CdS/W interface in

Fig. 3e reduces
upon biasing (Fig. 3f).”.

20. In Page 9, Line 195, “As a result, the Schottky barrier collapses and an Ohmic contact is formed at the CdS/W interface.” is changed to “Therefore, the Schottky barrier at the CdS/W interface reduces.”.
21. In Page 9, Line 197, “Discussion” is added.
22. In Page 11, Line 230-236, we have added some discussions about the influence of electrochemical potential on electrically driven cation exchange reaction.
23. In Page 11, Line 242-245, “Thirdly, the radius of the ion to be exchanged should not be much larger than that of Cd²⁺ to ensure sufficient room for ion movement. Ag⁺ radius (142 pm) is much larger than Cd²⁺ radius (92 pm), which means that Ag cannot replace Cd in CdS NW.” is changed to “As Ag₂S is a good ionic conductor as Cu₂S, the reaction may undergo similar to Cu₂S formation. However, the CE reaction for Ag electrode was not observed. This might be owing to the deficient electrochemical potential supply to ionize silver before CdS NW melts.”.
24. In Page 13, Line 289, “Our results open a new perspective to control the synthesis of nanocrystals at the atom level” is changed to “Our results open a new perspective to selective cation exchange on individual nanocrystals”.
25. In Page 13-15, Line 293-324, we have added some contents about experiment methods.
26. Green in all figures was changed to indigo to avoid confusion for colour-blind readers.
27. We added Figure S3, Figure S5, Figure S6 and Figure S8 and

deleted some contents about experiment methods in supporting information.

28. We have corrected the format of references followed by Nature guideline.
29. All figures were reordered in the revised manuscript and the revised supporting information.

REVIEWERS' COMMENTS:

Reviewer #1 (Remarks to the Author):

Overall I think that the replies of the authors are satisfactory, I have no other major concern.

Reviewer #2 (Remarks to the Author):

From the revised manuscript as well as response letter, I believe the authors clarified all the quote that the reviewer suggested. In particular, it's great to add detailed description of experimental setup and the formation mechanism that authors discuss is now highly convincing. The final product looks very good and is recommended to publish in Nature Communications.

Reviewer #3 (Remarks to the Author):

After reviewing the authors' responses and edits to the original manuscript, I feel they have adequately addressed all the concerns that were raised by the referees.